# Gene Expression Divergence in *Eugenia uniflora* Highlights Adaptation across Contrasting Atlantic Forest Ecosystems

**DOI:** 10.3390/plants13192719

**Published:** 2024-09-28

**Authors:** Andreia C. Turchetto-Zolet, Fabiano Salgueiro, Frank Guzman, Nicole M. Vetö, Nureyev F. Rodrigues, Natalia Balbinott, Marcia Margis-Pinheiro, Rogerio Margis

**Affiliations:** 1Programa de Pós-Graduação em Genética e Biologia Molecular, Departamento de Genética, Instituto de Biociências, Universidade Federal do Rio Grande do Sul (UFRGS), Porto Alegre 90010-150, Brazil; aturchetto@gmail.com (A.C.T.-Z.); nicolevetors@gmail.com (N.M.V.); nati.balbinott@gmail.com (N.B.); marcia.margis@ufrgs.br (M.M.-P.); 2Laboratório de Biodiversidade e Evolução Molecular, Departamento de Botânica, Universidade Federal do Estado do Rio de Janeiro, Rio de Janeiro 20551-030, Brazil; fabiano.salgueiro@unirio.br; 3Facultad de Medicina, Universidad Científica del Sur, Lima 15307, Peru; frank.guzman.e@gmail.com; 4Departamento de Biofísica, Universidade Federal do Rio Grande do Sul (UFRGS), Porto Alegre 90010-150, Brazil; 5Centro de Biotecnologia, Universidade Federal do Rio Grande do Sul (UFRGS), Porto Alegre 90010-150, Brazil

**Keywords:** Neotropics, Atlantic Forest, abiotic stress, local adaptation, Myrteae tribe, environmental changes

## Abstract

Understanding the evolution and the effect of plasticity in plant responses to environmental changes is crucial to combat global climate change. It is particularly interesting in species that survive in distinct environments, such as *Eugenia uniflora,* which thrives in contrasting ecosystems within the Atlantic Forest (AF). In this study, we combined transcriptome analyses of plants growing in nature (Restinga and Riparian Forest) with greenhouse experiments to unveil the DEGs within and among adaptively divergent populations of *E. uniflora*. We compared global gene expression among plants from two distinct ecological niches. We found many differentially expressed genes between the two populations in natural and greenhouse-cultivated environments. The changes in how genes are expressed may be related to the species’ ability to adapt to specific environmental conditions. The main difference in gene expression was observed when plants from Restinga were compared with their offspring cultivated in greenhouses, suggesting that there are distinct selection pressures underlying the local environmental and ecological factors of each Restinga and Riparian Forest ecosystem. Many of these genes engage in the stress response, such as water and nutrient transport, temperature, light intensity, and gene regulation. The stress-responsive genes we found are potential genes for selection in these populations. These findings revealed the adaptive potential of *E. uniflora* and contributed to our understanding of the role of gene expression reprogramming in plant evolution and niche adaptation.

## 1. Introduction

Understanding the mechanisms underlying the ability of organisms to adapt to different environmental conditions provides insights into the forces that allow populations and species to respond to environmental challenges. Gene expression and regulation are fundamental molecular mechanisms underlying plastic and evolutionary responses [1]. Since the 1970s, it has been proposed that many adaptations may have arisen from changes in gene regulation since molecular and morphological data have indicated that protein divergence is insufficient to explain the extensive phenotypic variation observed between and within species [2,3]. Since then, several studies have recognized variation in gene expression as a significant drive of phenotypic evolution, enabling adaptation within and between species to their native habitats [1,4,5,6,7,8]. As gene expression connects genotype to cellular and organismal physiology and potentially adaptive phenotypes, plasticity in gene expression can serve as a functional link in response to changing environments within and across generational timescales [9,10]. Therefore, investigating the transcriptional variation among natural populations living in distinct environments can contribute to our knowledge about species adaptation and diversification. Furthermore, these studies can contribute to understanding how species cope with climate change [11]. However, studies evaluating gene expression in non-model plant species are still limited.

In the last decade, studies in non-model species have been revolutionized by developing high-throughput sequencing technologies [12]. Using RNA-Seq data, measuring and comparing gene expression levels across species is possible, even when genomic sequences are unavailable [7,13]. RNA-Seq and transcriptomic analyses have revealed the molecular basis (both plastic and evolved) of physiological responses to environmental stressors. For instance, using transcriptome analysis of the Australian species *Banksia hookeriana*, Lim et al. [14] demonstrated the role of phenotypic variation and regulation of gene expression on the species’ capacity for rapid adaptation to climate change. They showed a correlation between differentiated phenotype and gene expression, indicating that the adaptive mechanism is heritable through natural selection or epigenetic processes. Studying adaptation to alpine environments, Wos et al. [15] used transcriptomic and phenotypic data to explore the evolution of ancestral plasticity during alpine colonization in *Arabidopsis arenosa*. When comparing gene expression between the foothill and alpine ecotypes, the authors showed that ancestral plasticity tended to be more reinforced than reversed during adaptation to an alpine environment. The role of gene expression in adaptive divergence and in determining ecologically based phenotypic differences among co-occurring species and their hybrids was demonstrated by Leal et al. [16]. They conducted a transcriptome analysis of sympatric and allopatric populations of two Orchidaceae that hybridize, *Epidendrum fulgens* and *E. puniceoluteum*, and showed that in sympatry, species exhibited differential expression in genes related to salt and waterlogging tolerance. The hybrid individuals showed a gene expression profile similar to flooding-tolerant *E. puniceoluteum*.

Widely distributed species with diverse and heterogeneous ecological habitats provide an excellent opportunity to explore the underlying molecular mechanisms of local adaptation. *Eugenia uniflora* L. (Myrtaceae) is an endemic and widely distributed species from the Atlantic Forest (AF) [17]. This Neotropical rainforest system comprises a mosaic of distinct ecosystems with contrasting characteristics and harbors one of the greatest biodiversities in the world [18,19]. *Eugenia uniflora* is popularly known as the Brazilian cherry or pitanga and can grow in different environments within the AF. The habitat range of this species exhibits significant variability in terms of precipitation, soil composition, and temperature preferences [20]. In its native habitat of the Atlantic Forest, *E. uniflora* can grow in contrasting ecosystems such as Restinga (RE) and Riparian Forests (RFs). These two ecosystems are different in location, vegetation, and environmental conditions. Riparian Forests, known as gallery forests, are located along freshwater bodies with rich and moist soil. At the same time, Restingas are coastal, dry, sandy ecosystems with salt- and nutrient-poor conditions. This species exhibits a remarkable phenotypic variation throughout its distribution range (Figure 1) [20]. This variation in morphological traits can be attributed to phenotypic plasticity or evolutionary changes. Riparian Forest individuals—subjected to an ombrophilous forest—are woody trees, growing up to ten meters tall, while the ones in Restinga—subject to sandy and nutrient-deficient coastal soil, high insolation, strong wind currents, and considerable temperature changes—are bushes, growing up to no more than three meters tall. *Eugenia uniflora* also displays leaf shape and size differences according to its occurrence in different environments. Leaves from Restinga plants are larger, with a rounded base and obtuse apex, while Riparian Forest plants have smaller leaves with an attenuated base and an acute apex [21]. These contrasting environments can drive local adaptation and consequently promote diversification.

Previous studies with Eugenia uniflora have demonstrated that populations inhabiting these distinct environments are quite different. A phylogeographic study of *Eugenia uniflora* based on plastidial markers revealed high population structure and lineage divergence associated with the phytogeographical changes in the Atlantic Forest [17]. An SNP-based study has identified distinct population structures of *E. uniflora* within the Atlantic Forest and genetic and phenotypic signals of local adaptation [20]. Interestingly, studies of both plastidial and nuclear (SNP) markers showed very low genetic diversity in populations associated with Restinga environments. Studying the P5CS gene and proline biosynthesis in *E. uniflora*, Anton et al. [22] found differences in the proline accumulation and P5CS gene expression of plants from Restinga and Riparian Forests under growth-controlled conditions. In addition, genome-wide studies identified MYB (v-myb avian myeloblastosis viral oncogene homolog) [23] and DOF (DNA-binding with one finger) [24] transcription factor (TF) gene families in E. uniflora. These TFs play crucial roles in regulating gene expression in response to biotic and abiotic stresses in plants. Both MYB and DOF genes were differentially expressed under drought stress, being potential genes involved in adaptation to diverse environmental conditions. Although some pieces of evidence from previous studies have helped us understand the evolutionary dynamics of this species, there are still many unsolved questions that need to be answered to assemble this evolutionary puzzle. Moreover, the molecular evolution of constitutive and plastic expression divergence across contrasting native environments remains unexplored.

In this study, we investigated the variation in gene expression between two contrasting populations of *Eugenia uniflora*: Restinga and Riparian Forest (Figure 1). We compared transcript levels of plants growing in the wild and greenhouse individuals exposed to the same controlled edaphoclimatic conditions to address the following primary questions: (i) Are there differences in the gene expression profiles between *E. uniflora* individuals from Restinga and Riparian Forest ecosystems? (ii) Which group of genes are differentially expressed in response to specific local environmental conditions? (iii) Is it a product of evolution in gene regulation or a shared ancestral plasticity?

## 2. Results

### 2.1. Comparing Samples from Different Environments

A principal component analysis (PCA) of the samples created a pattern that clearly shows the proximity of the three related biological samples from each of the four experimental groups: in situ Restinga (REn) and Riparian Forest (RFn), and Restinga (REc) and Riparian Forest (RFc) seeds grown in greenhouses (Figure 2A). A similar grouping pattern can be observed in the dendrogram (Figure 2B), where the REn samples are clustered as the most external group.

### 2.2. Patterns of Differentially Expressed Genes (DEGs)

Samples collected in contrasting natural environments presented 2122 differentially expressed genes, one of the highest levels among the comparisons (Figure 2C). When the pattern of gene expression of plants growing in the wild was compared with their offspring from the same origin but cultivated in greenhouses, the values dropped to 1650 in the case of Riparian Forest (RFc versus RFn) but showed a slight increase to 2378 for Restinga (REn versus REc). An even greater difference was observed when REn was compared to RFc (2999), while the lowest number of differentially expressed genes was observed between samples from REc and RFn (Figure 2C).

Individuals from the same group present quite large intrinsic variability in gene expression but cluster together when differential gene expression is observed in more detail as a heatmap (Figure 3A).

We split and defined the differential expression patterns among the four groups in 10 profiles, namely Baaa, BaaB, aaaB, BBaa, aaBB, BBBa, aBBa, aBBB, a(aB)(ab)B, and B(aB)(aB)a, where “B” indicates high expression and “a” corresponds to lower relative expression levels, always respecting the RFn, RFc, REc, and REn group order (Figure 3B). The profiles a(aB)(ab)B and B(aB)(aB)a correspond to no differences among REn vs. REc and RFn vs. RFc and are not indicated in the boxplots. A total of 1232 genes follow these ten common relative induction or repression patterns, ranging from 37 to 311 genes depending on the cluster. A complementary analysis of DEGs among the four groups is also presented in a Venn diagram (Appendix A).

Considering samples grown in the greenhouse (REc and RFc) as a calibration scenario, we observed a substantial difference in gene expression. More specifically, among the samples from Restinga, 311 genes showed increased expression (aaaB), while 245 genes exhibited decreased expression (BBBa). Conversely, in Riparian Forest samples, only 42 genes demonstrated increased expression (Baaa), and 40 genes displayed decreased expression (aBBB).

### 2.3. Genes Modulated by the Environment Compared to Those Exhibiting Constitutive Expression within Populations

The expression pattern profiles aaBB (37) and BBaa (81) reveal genes whose expression is constitutively modulated in samples sharing a common origin, even when environments were altered from natural to greenhouse conditions (Figure 3B).

The contrasting aaaB (311) and Baaa (42) encompass genes that are constitutively upregulated in the Restinga (REn) and Riparian Forest (RFn), respectively, as these profiles correspond to samples from adult trees continuously exposed to the specific ensemble of their natural biotic and abiotic stresses.

Representative genes related to adaptive mechanisms in Restinga (Figure 4C) or Riparian Forest (Figure 4D) and those constitutively modulated (Figure 4A,B) cover pathways and mechanisms related to water and nutrient transport, response to light, temperature, and oxidative stress, as well as those associated with secondary metabolite synthesis and transcription factors. Only a set of representative genes were indicated in Figure 4. A list of all genes with altered expression among groups, associated with biological functions and identification, and clustered according to their relation to water and transporters, response to high temperature and protein turnover, adaptation to different light intensities, genes related to oxidative stress, secondary metabolites, transcription factors, and hormones can be found in Appendix A.

## 3. Discussion

It is well known that genetic diversity is one way that underpins long-term adaptive ability, and a decline in genetic diversity often results in loss of fitness or even population extinction [25,26]. However, adaptation is not solely reliant on changes in DNA coding sequence. Gene expression diversity enables population persistence in changing environments and plays a vital role in phenotypic plasticity and adaptation [27]. Some recent studies have shown the contribution of gene expression in plant adaptation [28,29]. However, due to the lack of studies on the role of gene expression in adaptation, we are only beginning to understand how regulatory mechanisms contribute to the adaptive divergence of populations. In addition, studies on gene expression in native plant species are challenging since natural populations can present high variability. In the present study, we used RNA-Seq data from two contrasting populations of *Eugenia uniflora* to explore the role of gene expression variation underlying the ability of this species to inhabit challenging environments in the Atlantic rainforest ecoregion. Previous studies showed that populations of *E. uniflora* associated with Restinga have lower genetic diversity than populations associated with forests [17,20]. Here, we showed that these two different populations of *E. uniflora* are remarkably distinct at the gene expression level, both when plants from the natural environment and the offspring of plants of the same origin cultivated in greenhouses were compared. This result suggests that plastic and genetic differences in gene expression are important to maintaining populations in these distinct environments. Many gene expression differences appear to be associated with molecular mechanisms of adaptation, suggesting that the variation in gene expression may contribute to the adaptive capacity of this species. Equivalent results were found for Neotropical orchids *Epidendrum fulgens* and *E. puniceoluteum* from the Brazilian Restingas. In these two species, much of the variation in gene expression appears to be associated with genetic adaptation mechanisms [16]. A study with dune-adapted prairie sunflowers using a common garden experiment also showed the effect of gene expression variation in divergent ecotypes of this species [29].

Here, we found that plants from the Restinga population exhibited the most divergent expression pattern, even when plants from the natural environment were compared to their offspring grown in the greenhouse. We found a higher number of differentially expressed genes when plants from Restinga were compared with their offspring cultivated in greenhouses (REn vs. REc) than those from Riparian Forest and their offspring (RFn vs. RFc) (Figure 3). This variation in gene expression can be a result of selection pressures underlying the edaphoclimatic and ecological differences between the Restinga and Riparian Forest ecosystems. The Restinga ecosystem is an extreme environment with sandy soils poor in nutrients, high salinity, low water availability, high temperatures (daily/seasonal), and sea spray [30]. Conversely, Riparian Forest is defined by several distinct riverine-border plant communities and habitats [18,31], with variable access to sunlight, water, and flood regimes and soil rich in nutrients. In turn, gene expression analysis reveals pronounced distinctions among these populations in their adaptive responses to environmental stress. These differences will be further discussed in light of environmental and ecological factors likely affecting gene expression patterns in heterogeneous habitats within the Atlantic Forest.

### 3.1. Water and Nutrient Transport

The transport of water and nutrients is a critical physiological process for plant growth, development, and survival. The ability of plants to regulate this process is essential for tolerating challenging environments and adapting to new environmental conditions. This is particularly important for the two populations of *Eugenia uniflora* studied here, which grow in environments that differ in water and nutrient availability [20]. Thus, we expect that individuals from each environment present distinct mechanisms to tolerate and adapt to each condition. Our results showed several differentially expressed genes related to water and nutrient transport, such as aquaporin (PIP1.3), metal transport (YSL3), iron transporter (IRT), ammonium transporters (AMTs), nitrate transporters (NRTs), sulfate transporter, and glutamine synthase (GS) (Figure 4 and Figure 5, Appendix A). Aquaporins mainly mediate cellular water movement, facilitating the passive exchange of water across membranes [32,33]. In addition to water, these proteins may also conduct small neutral molecules and gases, including carbon dioxide (CO_2_) and oxygen (O_2_) [34,35,36]. We found a member of the aquaporin family (PIP1-related) highly expressed in *E. uniflora* plants growing in the natural environment of Restinga (REn). It has been demonstrated that plants overexpressing PIP1 exhibited changes in water use efficiency (WUE), increasing osmotic water permeability and resistance to drought and salinity [37,38]. These findings suggest that these proteins could be involved in the mechanisms of adaptation of plants of *E. uniflora* to persist in Restinga. Interestingly, another PIP aquaporin (PIP2-related) had lower expression in these same plants compared both with their offspring grown in greenhouse conditions and with Riparian Forest plants, either from natural environments or their offspring (Appendix A). Previous studies have shown that the knocking out of PIP2;3 had an impact on the expression of other PIP genes, causing the significant upregulation of other PIPs, like PIP1;3 [39]. This suggests a compensatory upregulation of PIP could be taking place because of the environmental conditions *E. uniflora* plants face in Restinga. Another possible scenario could be the distinct gene expression patterns of PIP isoforms in plant organs and tissues [40], as we analyzed only leaves from *E. uniflora* and aquaporins are known to act in the regulation of water uptake in roots.

Nitrogen transport and metabolism-related genes, like AMTs, NRTs, and GS, were also differentially expressed. Nitrogen is one of the essential macronutrients for plant growth, and ammonium (NH^4+^) and nitrate (NO^3−^) are the two primary inorganic nitrogen sources absorbed by plant roots [41]. The AMT3 and NRT2 genes, involved in ammonium and nitrate transport, were highly expressed in *E. uniflora* from Restinga (natural—REn). In *Arabidopsis thaliana*, it was demonstrated that NRT2.4, one of seven NRT2 gene family members, had its expression induced under N starvation [42]. As one strategy to tolerate the sandy soil naturally low in nitrogen found in the Restinga ecosystem, plants can adjust by increasing their levels of nitrogen transporters, as evidenced by our results in *E. uniflora* growing in Restinga. The overexpression of AMT1;1, an ammonium transporter from *Puccinellia tenuiflora,* promoted early root growth after seed germination in transgenic Arabidopsis under salt stress conditions [43]. These findings suggest that ammonium transport relieves ammonia toxicity caused by salt stress. This mechanism can be crucial to plants from Restinga to tolerate the high salinity they are exposed to. Interestingly, two glutamine synthetase (GS1) genes were also upregulated in plants from Restinga in natural conditions (Appendix A). This enzyme is crucial for nitrogen metabolism, and its higher expression level could result in increased nitrogen assimilation in plants exposed to the nitrogen-poor soils of Restinga [44]. The offspring of plants from Restinga restored the expression levels of AMTs, NRT, and GS1 when growing in nutrient-sufficient conditions, suggesting the plasticity of gene expression in dealing with environmental conditions.

### 3.2. Responses to High Temperatures

Plants, being immobile organisms, face inherent challenges in adapting to environmental shifts, resulting in the development of intricate regulatory systems to fight the stresses associated with high temperatures, such as those observed in Restinga. Appendix A presents a list of 57 genes associated with responses to high temperature, and 48 of them (84%) were upregulated in individuals from Restinga. Among them, the ubiquitin proteasome pathway stands out, employing E3 ligases to mark proteins for degradation via the 26S proteasome. These E3 ligases, grouped into four main structural categories, play crucial roles in diverse biological processes within plants, encompassing DNA repair, photomorphogenesis, phytohormone signaling, and responses to biotic stress. Notably, a significant portion of E3 ligase targets are proteins crucial for responding to various abiotic stresses, not only temperature, but also salt, drought, and cold [45]. The E3 ligases and their associated substrates are specifically linked to abiotic stress and to the chaperone network of proteins [46]. In our results, E3-ubiquitin ligases from the ATL31, CHIP, MARCH, and RGLG families were upregulated in samples growing in the Restinga area (Figure 5 and Appendix A). This could be part of the adaptive response of these plants to cope with the harsh environmental conditions of Restinga, either by specifically targeting proteins and fine-tuning stress responses or by regulating global protein turnover and maintaining cellular homeostasis.

In response to challenging environmental conditions, plants accumulate specific stress-responsive proteins, notably heat-shock proteins (HSPs) and late embryogenesis-abundant (LEA) proteins (Figure 4C), as previously demonstrated in situations of salinity, extreme temperatures, and water stress [47]. These proteins serve as guardians, shielding cells during stress by preserving their functional structure [48,49]. Abiotic stresses disrupt the usual function of enzymes and proteins, making it crucial for cellular survival to prevent their aggregation and maintain their proper configurations [50].

Under severe environmental changes that lead to protein denaturation, the synthesis of HSP70s is triggered, acting as molecular chaperones to aid in a variety of cellular processes necessary to withstand stressful conditions [51]. Not only HSP70 (Figure 4C) but a series of other chaperones like HSP83, HSP90, heat-shock factors HSF24 and 32, the dnaJ homologs ANJ1, B3, B3, B9, and ERDJ3B, and the chaperones ClpB1 and B3 were induced in Restinga plants (Appendix A). In the challenging environment of Restinga, the increased expression of chaperones significantly contributes to plant adaptation and survival, as it enables plants to manage stress-induced protein denaturation and misfolding. Chaperones play a pivotal role in protein synthesis, maturation, degradation, and targeting during stressful conditions. Additionally, they stabilize proteins and membranes and facilitate protein refolding, ensuring the proper functioning of proteins even in the face of misfolding or inactivity. Due to their importance in actively maintaining the integrity of the cellular proteome, chaperones are found in key cellular compartments like the cytosol, mitochondria, and chloroplasts.

### 3.3. Managing Contrasting Light Intensities

Photosystem II (PSII) stands as one of the most vulnerable elements within the photosynthetic apparatus, withstanding the most abiotic stress. Alongside the generation of reactive oxygen species (ROS) prompted by such stressors, ROS can also emerge from the absorption of excessive sunlight by the light-harvesting complex. These ROS possess the capacity to harm the photosynthetic machinery, especially PSII, leading to photoinhibition caused by an imbalance in the photosynthetic redox signaling pathways and hindering the repair mechanisms of PSII. Native plants with enhanced tolerance to abiotic stress must thoroughly control ROS signaling, and the regulatory roles of diverse components, including protein kinases, transcription factors, and phytohormones, in the responses of the photosynthetic machinery to abiotic stress are imperative. In this context, the upregulation of genes linked to degradation does not imply senescence; rather, it signifies an association with an increased turnover of chloroplastic proteins in response to stressful conditions [52,53].

Plants from Restinga have increased expression of two important genes for maintaining PSII structure and turnover: Maintenance of PSII under high light (MPH1) and psbP, an extrinsic protein that can affect chlorophyll content and efficiency of photosynthesis, whose release from PSII due to the interaction with malondialdehyde produced during thermal stress has been demonstrated [54]. Different chloroplast proteinases encoding genes such as FTSH6 and FSTH9 implicated in protein turnover under stress conditions [55] are induced in Restinga and Riparian Forest groups. In this scenario, it is important to mention that pheophytinase transcripts are also increased in plants from Restinga. Curiously, another extrinsic protein of PSII, psbR, involved in oxygen evolution [56], had its expression decreased, but only in samples from natural environments, not those grown in the greenhouse (Figure 5 and Appendix A). In parallel, diverse chlorophyll-ab-associated proteins (Cab) are differentially expressed among the four groups. Besides genes related to plastid protein synthesis, the translation initiation factor (eIF1) and protein of 21kDa of large ribosomal subunit (RPL21) were constitutively induced in REn and REc (Figure 4 and Figure 5 and Appendix A).

Plants from Restinga also have increased transcript levels of key genes in the Calvin–Benson cycle, as exemplified by ribulose 1, 5-bisphosphate carboxylase/oxygenase (Rubisco) [57], and sedoheptulose-1,7-bisphosphatase [58], while in Riparian Forest, starch synthase is increased. There is an increase in the expression of genes involved in the synthesis of long-chain and saturated lipid compounds (fatty acid desaturase 4 and very-long-chain enoyl-CoA reductase), a clear response to the higher temperatures to which Restinga plants are subjected (Figure 4 and 5 and Appendix A).

### 3.4. Other Genes Modulated by Environmental Stresses

Genes involved in terpenoid biosynthesis were highly expressed in *Eugenia uniflora* plants from Riparian Forest. Terpenoids are a class of plant secondary metabolites involved in environmental adaptation and stress tolerance. As the largest class of natural products, terpenes can defend many species of plants against predators, pathogens, and competitors [59]. The higher level of expression of terpene synthase (TPS) and 2 (-)-germacrene D synthase in plants from Riparian Forest (Figure 4 and Figure 5 and Appendix A) can be due to the fact that plants associated with forests may be more exposed to attack by predators, pathogens, and competitors than those from Restinga. TPS is the main enzyme in terpenoid biosynthesis and can use multiple substrates to produce a variety of terpenoids. It was demonstrated that substrate preference and terpene product profiles may vary in response to environmental fluctuations [60]. Studying terpenoid emission and expression of TPS genes in rice, Yuan et al. [61] identified three TPS genes responsible for the production of the majority of insect-induced volatiles, two of them encoding sesquiterpene synthases. The two TPS genes highly expressed in Riparian Forest plants encode sesquiterpene synthases. Sesquiterpenes accumulated following herbivore attacks, involved in the defense against the herbivores or attracting natural enemies to fight them [62,63]. The expression of the sesquiterpene β-caryophyllene gene is upregulated in *E. grandis* in response to *C. austroafricana* infection [64]. β-caryophyllene extract from the roots and pines of *Pinus halepensis* inhibited herbaceous plant growth [65].

Other noteworthy genes displaying heightened transcript levels are the pyridoxal 5′-phosphate synthase family (PDX1). These genes represent the conclusive and regulatory phase of vitamin B6 metabolism and have been linked to salt tolerance through the equilibrium maintenance of ROS (reactive oxygen species) and abscisic acid levels in plants [66]. Regarding genes associated with responses to oxidative stress and the redox state of proteins, it is observed that in Restinga plants, there is an increase in catalase, glutathione S-transferase (GST), and glutaredoxin transcripts, whereas different thioredoxins and cytochrome P450 are induced in both groups of plants in the field environment, while alcohol dehydrogenase (ADH) is induced in plants from Riparian Forest and aldehyde oxidase (GLOX1) is reduced in Restinga (Figure 4 and Figure 5; Appendix A).

Salt is a severe environmental stressor that affects the growth and development of plants. In Restinga areas, plants are affected by saltwater intrusion due to their proximity to the ocean and also periodic salt spray and tidal inundation. WRKY51 is one of the transcription factors with a higher increase in *Eugenia uniflora* from Restinga, more than 10-fold compared to the Riparian Forest. The overexpression of the poplar homolog WRKY51 in *Arabidopsis* improved salt tolerance in comparison to the more sensitive phenotype in *wrky51*-knockout mutants [67]. Numerous other members of the WRKY gene family have been associated with responses to abiotic stresses [68]. Therefore, it is not unexpected that WRKY-1, 65, and 75 have also been induced and that no WRKY gene was modulated in the plants of the Riparian Forest. Many other transcription factors and receptors had their transcript levels affected (DREB, Myb, NAC, BRI1, LRR), but with a challenging correlation in the sampled groups. A final observation concerns the NFY-A and SOC1 genes, which exhibited increased expression exclusively in samples collected from natural environments, not those cultivated in the greenhouse (Figure 5 and Appendix A). Both transcription factors are crucial in responding to and tolerating multiple abiotic stresses. They collaborate by forming heterodimer complexes, working in concert to enhance the plant’s resilience [69,70]. Two recent studies have shown that MYBs and DOF transcription factors are differentially expressed in *Eugenia uniflora* under drought stress [23,24]. This demonstrates that TFs can be involved in *E. uniflora* adaptation to local environments.

## 4. Materials and Methods

### 4.1. Study System and Experimental Design

Two populations of *Eugenia uniflora* were selected for this study based on previous studies [17,20] (Figure 1). These populations are located in two distinct environments of the species occurrence in the Atlantic Forest. Distinct soil types, temperature ranges, and precipitation patterns characterize these sample sites. One sample site (RE) is located in the Restinga ecosystem in the Rio Janeiro state (22°56′09.16′′S; 42°19′25.20′′W), characterized by nutrient-poor and sandy soils, high salinity, and limited water availability [30]. The other sample site (RF) is in the Riparian Forest ecosystem in the Rio Grande do Sul state (27°27′13.99″S; 53°28′9.01″W), characterized by fertile soil with abundant water [18]. We collected 8 to 10 healthy young leaves of three individuals, located more than 20 m apart from each other, from each population for RNA extraction. The leaves collected in the field were snap-frozen in liquid nitrogen and later transported to the Federal University of Rio Grande do Sul (UFRGS) on dry ice and then stored at −80 °C until RNA isolation. The sample collection from each population was performed in November 2016.

*Eugenia uniflora* seeds were also collected from both populations and germinated in a greenhouse under controlled edaphoclimatic conditions (28 °C, 68% relative humidity) (REc and RFc). Leaves from six-month-old plants (three individuals from each site) were collected and immediately stored at −80 °C for RNA isolation. The RNA from both field and greenhouse samples was used for library construction and deep sequencing. We also collected leaves of an *E. uniflora* individual grown in an orchard at UFRGS for genome sequencing and assembly. We chose this individual because it is the same individual used for the reference transcriptome assembly [71] and the paired-end library (insert length of 250 bp) [72].

### 4.2. RNA Extraction, Library Preparation, and Deep Sequencing

Healthy leaves from different regions of the canopy of each tree in natural environments or from plants grown in the greenhouse were selected, and between 8 and 10 leaves were ground together in liquid nitrogen for subsequent total RNA extraction. RNA was isolated from the powdered leaves following a sequential protocol, using the CTAB extraction method [73], followed by further purification with the commercial kit Direct-zol™ (Zymo Research, R2050). RNA integrity was evaluated by electrophoresis on a 1% agarose gel and stained using GelRed^®^ (Biotium), then visualized under ultraviolet light. The concentration of RNA samples was checked in a Nanodrop Lite (Thermo Scientific, Waltham, MA USA). The RNA samples were sent to Macrogen Inc., Seoul, Republic of Korea, for messenger RNA enrichment, RNA fragmentation, cDNA synthesis, adapter ligation, PCR amplification, and sequencing. A total of 12 libraries were sequenced to produce paired-end reads of 100 bp read length using an Illumina Hiseq 4000. The genome of *E. uniflora* deposited in the NCBI Sequence Read Archive at Bioproject PRJNA784246 was crucial to improve the transcriptome assembly that was submitted to NCBI Sequence Read Archive at Bioproject PRJNA1121623 and PRJNA549455.

### 4.3. Read Trimming, De Novo Transcriptome Assembly, and Quality Assessment

The FASTQ files for each library containing the raw reads of cDNA libraries from the field and the greenhouse datasets were cleaned using the FASTQC [74] quality control tool and Trimgalore! (https://www.bioinformatics.babraham.ac.uk/projects/trim_galore/, accessed on 1 March 2021). All 5′ and 3′ adapter fragments, as well as ambiguous reads containing >5% unknown nucleotides (‘n’) and low-quality reads with more than 20% Q < 30 bases, were removed to ensure the accuracy of de novo assembly and subsequent analyses. The cleaned reads were aligned against the assembled genome (Bioproject PRJNA784246) with a similarity of 95% with the other parameters set to their default values (Appendix A). It is important to note that relatively low values of paired-end reads aligned to the genome were expected and can be explained by the fact that these are RNA samples obtained from native plants, from different populations, with a high degree of polymorphisms compared to the genome used. A list of annotated genes of *Eugenia uniflora* is available in Appendix A.

### 4.4. Differential Expression and Statistical Analysis

To explore and compare *Eugenia uniflora* responses to environmental locations at the transcript level, differential expression analysis was conducted using SARtools with DESeq2 [75], and *p*-values were adjusted with Benjamini and Hochberg’s correction to reduce the false discovery rate [76]. Comparisons were carried out between and within the samples living in the natural environments of Restinga (REn) and Riparian Forest (RFn) and grown in controlled greenhouse conditions (REc and RFc). To maximize the identification of genes involved in stress responses, genes with *p* < 0.05 were defined as differentially expressed, with no threshold set for fold change (FC). Genes were defined as up- or downregulated according to multiple pairwise comparisons: REn vs. RFn; REn vs. REc; RFn vs. REc, and REc vs. RFc. The statistical differences among samples were assigned with “a” to indicate downregulation or relative reduced expression and “B” to indicate upregulation or relative induced expression. An expression pattern table (Figure 3) and a Venn diagram (Appendix A) were constructed using genes that fit these criteria. The list of genes is in Appendix A.

## 5. Conclusions

The current study unequivocally highlights that a multitude of genes across various pathways were adjusted to facilitate adaptive resilience in *E. uniflora* across the Restinga and Riparian Forest ecosystems. These adaptive mechanisms consist of genes modulated in response to environmental stresses, as well as others differentially expressed constitutively among individuals from each ecosystem. This demonstrates how challenging it is to interfere with or select a specific group of genes in pursuit of greater adaptive potential under natural conditions. Thus, in the current perspectives of climate change, the premise remains valid that we should preserve a greater genotypic diversity, with an emphasis on diversities arising from geographically distinct populations and potentially contrasting environments. In addition, these findings support the idea that conservation efforts should also focus on preserving intraspecific diversity, especially in ecosystems that are threatened and vulnerable to climate change, such as the Atlantic Forest. By conserving this diversity, it is possible to maintain the full spectrum of adaptive potential and increase the chance of species survival in a rapidly changing world.

## Figures and Tables

**Figure 1 plants-13-02719-f001:**
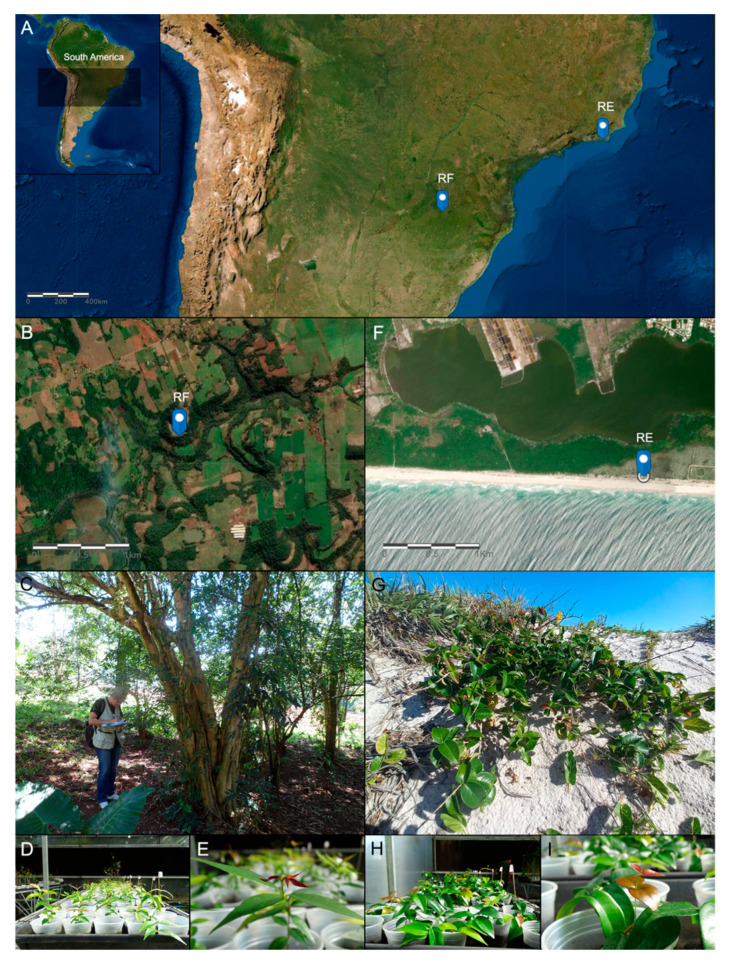
*Eugenia uniflora* distribution in the Atlantic Forest (AF), showing the locations where RNA was sampled for this study. (**A**) Map with the population sampled in the Restinga (RE) and Riparian Forest (RF) ecosystems within AF. (**B**) Satellite image showing the characteristics of the Riparian Forest ecosystem. (**C**) *E. uniflora* tree growing in the RF ecosystem. (**D**,**E**) *E. uniflora* from the RF ecosystem growing in the greenhouse. (**F**) Satellite image showing the characteristics of the Restinga ecosystem. (**G**) *E. uniflora* shrub growing in the RE ecosystem. (**H**,**I**) *E. uniflora* from the RE ecosystem growing in the greenhouse.

**Figure 2 plants-13-02719-f002:**
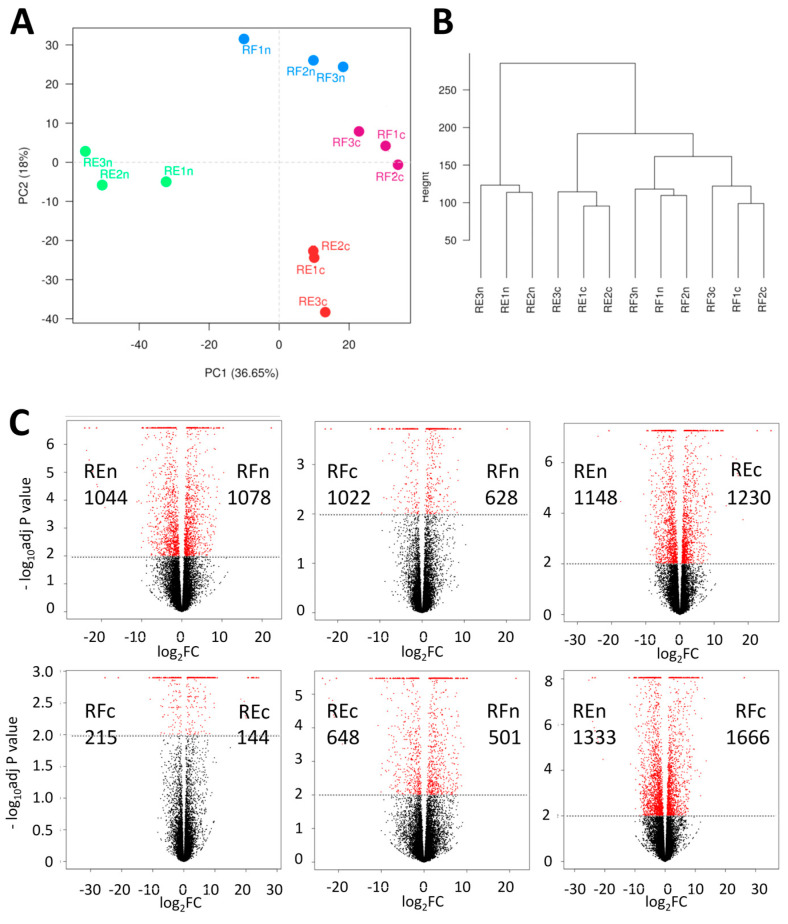
Sample clustering and the pattern and number of differentially expressed genes. (**A**) Principal component analysis of variation among samples (PCA) and (**B**) dendrogram showing the relation among all 12 samples from the four experimental groups (RFn and REn: Riparian Forest and Restinga staples collected in nature; RFc and REc: Riparian Forest and Restinga individuals originated from these areas but cultivated in the greenhouse). (**C**) Volcano plot showing the number of differentially expressed genes (in red) between six experimental groups with pair comparisons.

**Figure 3 plants-13-02719-f003:**
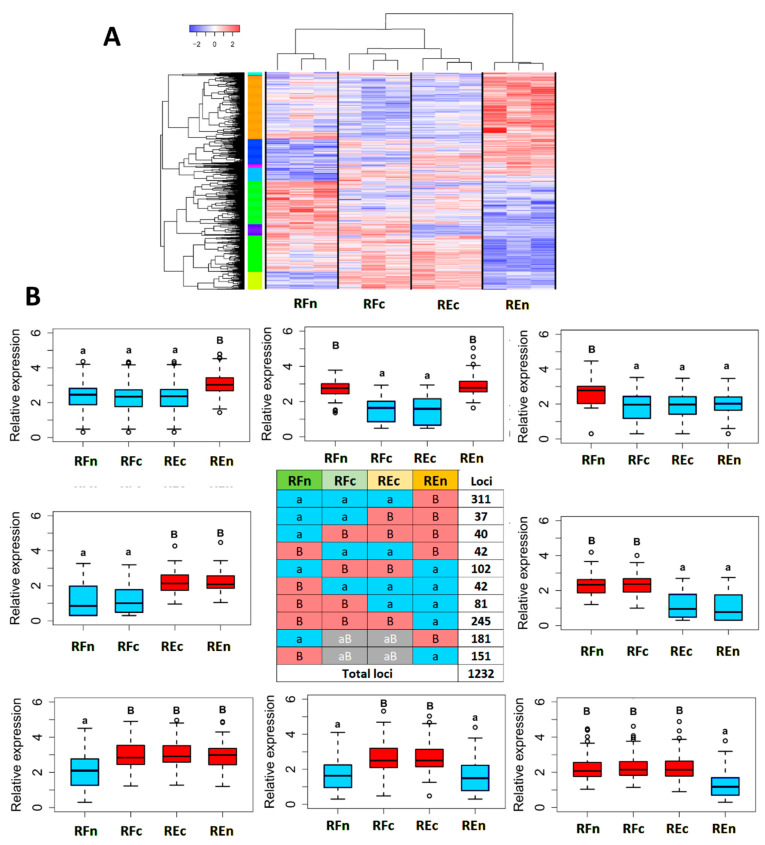
Differential gene expression patterns between sample groups. (**A**) Dendrogram and heatmap of differentially expressed genes among all three biological replicates of each of the four groups, where red marks correspond to genes with induced expression and blue marks to repressed ones. (**B**) Boxplots of the different expression patterns among the four groups. Statistical differences among samples are marked with “a” to indicate downregulation or relative reduced expression and by “B” to indicate upregulation or relative induced expression. The central table indicates the number of loci corresponding to each of the 10 expression patterns. The blue boxes correspond to repressed genes, and the red ones correspond to induced genes in comparative analyses and expression patterns.

**Figure 4 plants-13-02719-f004:**
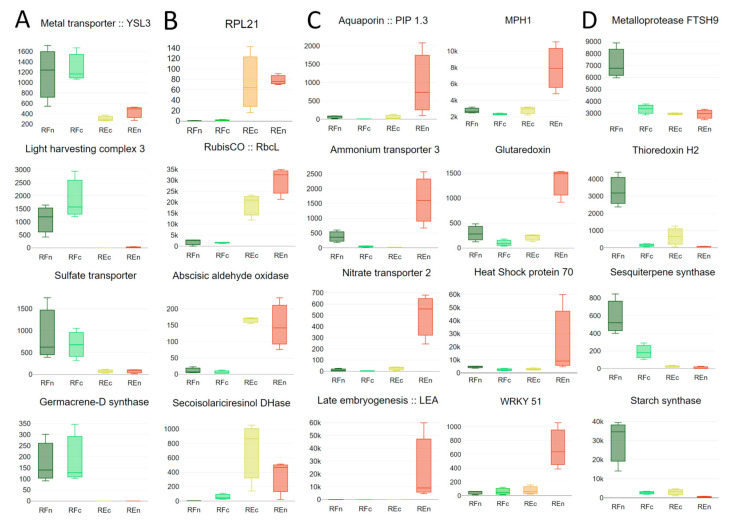
Histograms of relative gene expression among samples from the Riparian Forest collected in the field (RFn) or grown in greenhouses (RFc) and from Restinga collected in the field (REn) or in greenhouses. A set of representative genes constitutively modulated (**A**,**B**) or induced in Restinga and Riparian Forest field conditions, respectively (**C**,**D**).

**Figure 5 plants-13-02719-f005:**
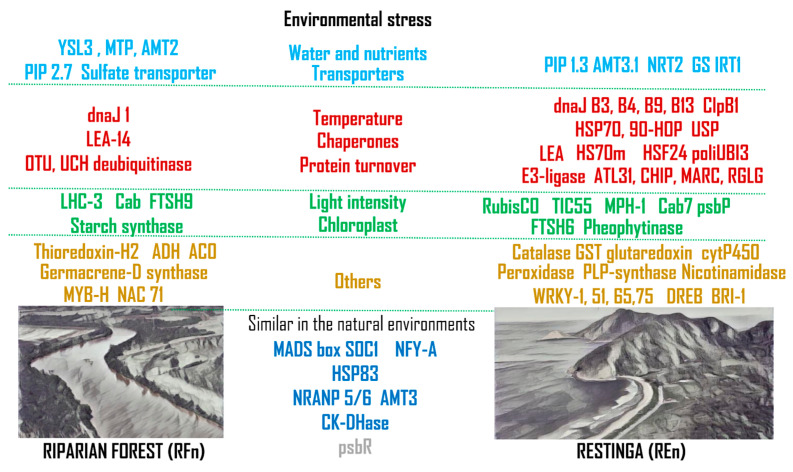
Schematic view of different abiotic environment stresses or factors and genes associated with mechanisms and pathways whose expression was increased in *Eugenia uniflora* growing in contrasting Riparian Forest (**left**) or in Restinga (**right**) ecosystems. Genes associated with heat stress, protein stability, and turnover are indicated in red; those related to light response and chloroplast homeostasis are shown in green; and genes involved in oxidative stress and transcriptional regulation are marked in gold. Genes modulated in both natural environments compared to greenhouse are also indicated in the last central column, in blue..

## Data Availability

The raw transcriptome datasets of *Eugenia uniflora* were deposited in the NCBI Sequence Read Archive at Bioproject PRJNA784246. The list of all annotated *E. uniflora* loci and CDS sequences was made available in Appendix A.

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
