# Peer review of "Gene Expression Divergence in Eugenia uniflora Highlights Adaptation across Contrasting Atlantic Forest Ecosystems"

_plants, 2024, doi:10.3390/plants13192719_

Round 1

Reviewer 1 Report

Comments and Suggestions for Authors

 I believe that the manuscript needs Minor revisions.

Minor Comments:

  1. Title Clarity: Consider revising the title for clarity. It could be more direct in highlighting the main findings, such as focusing on specific gene expression changes or adaptive mechanisms in contrasting environments.
  2. Abstract: The abstract would benefit from a clearer presentation of the key findings. The mention of specific genes or pathways involved in adaptation could strengthen the impact.
  3. Introduction:

·       The introduction effectively sets up the background, but it could be enhanced by providing a brief overview of the known adaptive strategies of Eugenia uniflora prior to this study. This would contextualize the novelty of your work.

·       The introduction refers to gene expression plasticity and adaptation but could briefly explain the significance of contrasting environments, particularly for readers less familiar with Restinga and Riparian ecosystems.

  1. Methods:

·       The RNA extraction methods are well-described. However, a more detailed explanation of the criteria for selecting plants from the two contrasting environments (Restinga and Riparian Forest) could be useful.

·       In the RNA-seq analysis, the use of SARtools with DESeq2 for differential expression analysis is appropriate, but the justification for the specific significance thresholds used (P < 0.05) should be provided.

  1. Figures:

·       Figures 2 and 3 are informative but could be made clearer with additional labels or annotations, particularly for non-expert readers who may not be familiar with the methods (e.g., PCA and heatmap analysis).

·       Figure 5 could include more descriptive labels to make it easier to understand the genes and pathways without referring back to the main text.

  1. Discussion:

·       The discussion thoroughly covers the implications of the findings, but it could benefit from a more explicit comparison with similar studies on adaptive gene expression in other plant species. This would help position your study within the broader context of plant adaptation research.

·       It would also be helpful to discuss any limitations or challenges in interpreting the gene expression data due to the variability inherent in natural populations.

  1. Conclusion: The conclusion could be strengthened by explicitly linking the findings to potential conservation efforts, particularly in preserving genetic diversity across distinct ecosystems like Restinga and Riparian forests.
  2. References: A few recent references in the field of gene expression and plant adaptation to climate change could be included to further support the study's relevance. Language and Style:

9.     The manuscript contains minor language issues, such as overly complex sentence structures in some sections. Simplifying these sentences can improve readability.

Comments on the Quality of English Language

Ensure that sentences are grammatically correct. Check for common issues like subject-verb agreement, correct tense usage, and appropriate article usage.

  • Ensure that ideas are expressed clearly and concisely. Avoid redundancy, and check if some sentences could be more straightforward.
  • Example: "Due to the fact that" could be replaced with "because."

Author Response

Reviewer 1

 The manuscript titled “Adaptive gene expression in two contrasting populations of Eugenia  uniflora L. (Myrtaceae) unveil its ability to persist in challenging environments.” This article addresses a significant topic in the field of evolutionary biology and plant adaptation by exploring the gene expression variation in Eugenia uniflora populations inhabiting contrasting environments. The focus on gene expression plasticity and its role in the species' ability to persist under diverse environmental stresses adds valuable insights into understanding the adaptive potential of non-model species, particularly in the context of global climate change. The study provides essential information regarding gene regulation mechanisms and environmental adaptation, which are crucial for biodiversity conservation and the prediction of species' responses to future climate conditions. I believe that the manuscript needs Minor revisions. 

Minor Comments:

  1. Title Clarity: Consider revising the title for clarity. It could be more direct in highlighting the main findings, such as focusing on specific gene expression changes or adaptive mechanisms in contrasting environments.

Answer: The authors modified the Title in order to make it more straightforward.

  1. Abstract: The abstract would benefit from a clearer presentation of the key findings. The mention of specific genes or pathways involved in adaptation could strengthen the impact.

Answer:  The main pathways, as indicated in figure 5, were mentioned in the abstract. Authors consider the list of individual genes to be inappropriate in the abstract. 

  1. Introduction: 
  • The introduction effectively sets up the background, but it could be enhanced by providing a brief overview of the known adaptive strategies of Eugenia uniflora prior to this study. This would contextualize the novelty of your work.
  • The introduction refers to gene expression plasticity and adaptation but could briefly explain the significance of contrasting environments, particularly for readers less familiar with Restinga and Riparian ecosystems.

Answer: This information was placed in the introduction, where both contrasting environments were detailed as well the adaptive characteristics of Eugenia uniflora in both places.

  1. Methods:
  • The RNA extraction methods are well-described. However, a more detailed explanation of the criteria for selecting plants from the two contrasting environments (Restinga and Riparian Forest) could be useful.

Answer: This information was placed in the introduction where both contrasting environments were detailed as well the adaptive characteristics of Eugenia uniflora in both places.

  • In the RNA-seq analysis, the use of SARtools with DESeq2 for differential expression analysis is appropriate, but the justification for the specific significance thresholds used (P < 0.05) should be provided.

Answer: A P<0.05 is an universal cutoff. Authors understand that the cutoff of fold change can circumvent two scenarios: i) a more stringent were only high variations were detected in detriment of the network identification. Also, it is known that several pathways are regulated by slight and pulse responses that cannot be detected as a strong fold change in expression; (ii) a lower fold change, but supported by statistical analyses, can create a better picture of the networks of interactions and pathways in response to external stimuli.

  1. Figures:
  • Figures 2 and 3 are informative but could be made clearer with additional labels or annotations, particularly for non-expert readers who may not be familiar with the methods (e.g., PCA and heatmap analysis). 

Answer:  The meaning of PCA and the information about heatmap were introduced in the legend of figures 2 and 3.

  • Figure 5 could include more descriptive labels to make it easier to understand the genes and pathways without referring back to the main text. 

Answer: Mention to each color label, corresponding to the different pathways, were added to the legend.

  1. Discussion:
  • The discussion thoroughly covers the implications of the findings, but it could benefit from a more explicit comparison with similar studies on adaptive gene expression in other plant species. This would help position your study within the broader context of plant adaptation research.
  • It would also be helpful to discuss any limitations or challenges in interpreting the gene expression data due to the variability inherent in natural populations. 

Answer:  We included some new studies that address the role of gene expression variation in plant adaptation. In addition, we highlighted the challenges in studying natural populations.

  1. Conclusion: The conclusion could be strengthened by explicitly linking the findings to potential conservation efforts, particularly in preserving genetic diversity across distinct ecosystems like Restinga and Riparian forests. 

Answer:  We added a sentence in the conclusion to explicitly link our findings with conservation efforts.

  1. References: A few recent references in the field of gene expression and plant adaptation to climate change could be included to further support the study's relevance.

Answer:  New references were added to the new version of the discussion.

 Language and Style: 

  1. The manuscript contains minor language issues, such as overly complex sentence structures in some sections. Simplifying these sentences can improve readability.  

Answer:  The text was checked for formatting and clarity

Reviewer 2 Report

Comments and Suggestions for Authors

The manuscript title “Adaptive gene expression in two contrasting populations of Eugenia uniflora L. (Myrtaceae) unveil its ability to persist in challenging environments” is conducted well and has scientific worth. I have some suggestions to improve the current version of the manuscript.

Summary of Manuscript:

Understanding plant responses to environmental changes is essential in the context of global climate change, particularly in species like Eugenia uniflora that thrive in diverse ecosystems within the Atlantic Forest. This study combined transcriptome analyses of E. uniflora from natural habitats (Restinga and Riparian Forest) with greenhouse experiments to explore gene expression variations among adaptively divergent populations. Significant differences in gene expression were identified between the two populations in both natural and greenhouse environments, indicating the species' adaptability to specific conditions. The most notable expression changes occurred when comparing Restinga plants with their greenhouse-grown offspring, highlighting distinct selection pressures in each ecosystem. Many differentially expressed genes were linked to stress responses, including water and nutrient transport, temperature regulation, light intensity, and gene regulation, suggesting their potential role in selection within these populations. The findings underscore the adaptive potential of E. uniflora and enhance our understanding of gene expression reprogramming in plant evolution and niche adaptation.

Comments for authors are as follows:

1.     Line 429-430: How do the authors justify the selection of the two contrasting populations of Eugenia uniflora for this study?Any specified reasons? What are the specific environmental challenges that the plant faces in each habitat?

2.     Can the authors provide more details on the experimental design, including the number of biological replicates? How many biological replicates were used please add this in methods section! and the specific conditions in the greenhouse experiment? Such as light intensity, light/dark cycle??

3.     How were the differentially expressed genes (DEGs) identified, and what statistical methods were used to validate these findings?

4.     The authors repeated this sentences twice in the manuscript (Results and methods section) “The statistical differences among samples were assigned with an “a” to indicate downregulation or relative reduced expression and by a “B” to indicate upregulation or relative induced expression.” I suggest authors to change the wording and provide all the statistical information with reference in a separated section under the methods for example “4.5. Statistical analysis”

5.     Line 146: In the method section “2.2.  Patterns of differential gene expression (DGE)” please use differentially expressed genes (DEGs) as in next line 147: the authors mentioned differentially expressed genes, so the heading should be consistent.

6.     Line 437: how many leaves, per plant per replication etc,, provide more details.

7.     First give a common title to figures 2, 3, etc., than explain the different subfigure such as A, B, C, etc…. like authors did in figure 1.

8.     Line 438: Changed this “liquid N2” to this “liquid nitrogen”

9.     The manuscript mentions the use of a commercial kit for RNA extraction but does not specify catalog number of the kit used. Used CTAB method or RNA kit or both? give more clarity to avoid confusion.

10.   It’s a suggestion! Consider adding a figure or a table summarizing the most relevant DEGs and their potential roles in stress response, which would aid in the reader's understanding of the key findings.

Author Response

 Reviewer 2

The manuscript title “Adaptive gene expression in two contrasting populations of Eugenia uniflora L. (Myrtaceae) unveil its ability to persist in challenging environments” is conducted well and has scientific worth. I have some suggestions to improve the current version of the manuscript.

Comments for authors are as follows:

  1.     Line 429-430: How do the authors justify the selection of the two contrasting populations of Eugenia uniflora for this study?Any specified reasons? What are the specific environmental challenges that the plant faces in each habitat?

Answer:  The justification and rationale for using these two population sets were provided in the introduction.

  1.     Can the authors provide more details on the experimental design, including the number of biological replicates? How many biological replicates were used please add this in methods section! and the specific conditions in the greenhouse experiment? Such as light intensity, light/dark cycle??

Answer:  These information were added to the previous information present in material and methods.

  1.     How were the differentially expressed genes (DEGs) identified, and what statistical methods were used to validate these findings?

Answer:  These analyses are explained in material and methods section 4.4.

  1.     The authors repeated this sentences twice in the manuscript (Results and methods section) “The statistical differences among samples were assigned with an “a” to indicate downregulation or relative reduced expression and by a “B” to indicate upregulation or relative induced expression.” I suggest authors to change the wording and provide all the statistical information with reference in a separated section under the methods for example “4.5. Statistical analysis”

Answer:  The subsection 4.4 title was modified to Differential expression and statistical analysis in order to clarify the information about the statistical analysis. Regarding the duplication of the text referencing the expression patterns, the authors chose to repeat the information in the legend of Figure 3 to make it fully self-explanatory. Therefore, the text was not repeated throughout the results section, but only in the legend of Figure 3, as justified, for clarity and ease of understanding.

  1.     Line 146: In the method section “2.2.  Patterns of differential gene expression (DGE)” please use differentially expressed genes (DEGs) as in next line 147: the authors mentioned differentially expressed genes, so the heading should be consistent. - DONE
  2.     Line 437: how many leaves, per plant per replication etc,, provide more details.  - DONE
  3.     First give a common title to figures 2, 3, etc., than explain the different subfigure such as A, B, C, etc…. like authors did in figure 1. 

Answer: A common and more general title was associated with figures 2 and 3.

  1.     Line 438: Changed this “liquid N2” to this “liquid nitrogen” - DONE
  2.     The manuscript mentions the use of a commercial kit for RNA extraction but does not specify catalog number of the kit used. Used CTAB method or RNA kit or both? give more clarity to avoid confusion.

Answer: The catalog number was added when the kit was mentioned in material and methods. Also, we clarify that the CTAB extraction was a pre-step made before the Direct-zol kit.

  1.   It’s a suggestion! Consider adding a figure or a table summarizing the most relevant DEGs and their potential roles in stress response, which would aid in the reader's understanding of the key findings.

Answer: Figure 5 contains a list of most relevant upregulated genes between natural environments that correspond to the main hubs of adaptive responses.

Reviewer 3 Report

Comments and Suggestions for Authors

18.9.2024

The manuscript entitled "Adaptive gene expression in two contrasting populations of Eugenia uniflora L. (Myrtaceae) unveil its ability to persist in challenging environments” was reviewed.

The manuscript delivers novel insights DEGs in Eugenia uniflora from two divergent environments. The manuscript delivers important finding, however, major data analysis are missing (Venn diagrams and qRT-PCR), in addition, an urgent update is required in cited articles. Therefore, I do not recommend the publication of this manuscript in "Plants" unless these corrections are made. Please see below for additional comments.

1.  General:

- The language is fine. But few weak expression and common mistakes need to be corrected. Here are JUST few examples:

- Line 18: replace “concerning” with “to combat”.

- Line 19: replace “occurs” with “thrives”.

- Line 22: replace “the expression variation” with “DEGs”.

- Line 430: replace “are in two” with “are located in two”.

- Line 434: replace “small water” with “limited water”.

- Line 436: replace “greater water availability” with “abundant water”.

- Line 440: replace “in each” with “from each”.

- Do not keep space between the temperature and the symbol, e.g. replace “28 °C” with “28°C”

2. Abstract:

- The abstract is missing any actual values of measurements and names of major novel DEGs; please you need to indicate some figures to support your conclusions and interpretations!

3. Introduction:

- Language is fine.

- Relatively long, please reduce!

- Figure 1 (G): it is inappropriate to show this image in highly ranked journal such as “Plants”. Please remove the lower part!

4. Results:

- Good presentation of data.

- Very informative supplementary files.

- Two major RNA-seq data analyses are MISSING:

1. Venn diagrams showing overlapping genes between the four major samples.

2. qRT-PCR to verify the RNA-seq data.

5. Discussion:

- Good, however incorporating more recent articles is recommended.

- Please you need to discuss your data with recent similar reports such as:

- Filgueiras, J. P. C., da Silveira, T. D., Kulcheski, F. R., & Turchetto-Zolet, A. C. (2024). Unraveling the Role of MYB Transcription Factors in Abiotic Stress Responses: An Integrative Approach in Eugenia uniflora L. Plant Molecular Biology Reporter, 1-12.

- Waschburger, E. L., Guzman, F., & Turchetto-Zolet, A. C. (2022). Genome-wide identification and analysis of dof gene family in Eugenia uniflora l.(Myrtaceae). Genes13(12), 2235.

- You need to discuss the possible epigenetic effect from original collection site on seedling raised in the greenhouse!

- You need to discuss the around 8000 cDNAs without annotation!

6. Materials and Methods:

- Lines 453-462: you do not need to indicate the detailed procedures as you cited the original articles.

- qRT-PCR to verify the RNA-seq data is missing.

7. References:

- It is recommended to incorporate more recent articles as only 15 out 71 (ca. 21%) of cited articles were published in the last five years.

Comments on the Quality of English Language

- The language is fine. But few weak expression and common mistakes need to be corrected. Here are JUST few examples:

- Line 18: replace “concerning” with “to combat”.

- Line 19: replace “occurs” with “thrives”.

- Line 22: replace “the expression variation” with “DEGs”.

- Line 430: replace “are in two” with “are located in two”.

- Line 434: replace “small water” with “limited water”.

- Line 436: replace “greater water availability” with “abundant water”.

- Line 440: replace “in each” with “from each”.

Author Response

 Referee3

The manuscript entitled "Adaptive gene expression in two contrasting populations of Eugenia uniflora L. (Myrtaceae) unveil its ability to persist in challenging environments” was reviewed.

The manuscript delivers novel insights DEGs in Eugenia uniflora from two divergent environments. The manuscript delivers important findings, however, major data analysis are missing (Venn diagrams and qRT-PCR), in addition, an urgent update is required in cited articles. Therefore, I do not recommend the publication of this manuscript in "Plants" unless these corrections are made. Please see below for additional comments.

  1. General:- The language is fine. But few weak expression and common mistakes need to be corrected. Here are JUST few examples:

- Line 18: replace “concerning” with “to combat”.  - DONE

- Line 19: replace “occurs” with “thrives”.  - DONE

- Line 22: replace “the expression variation” with “DEGs”.  - DONE

- Line 430: replace “are in two” with “are located in two”.  - DONE

- Line 434: replace “small water” with “limited water”.  - DONE

- Line 436: replace “greater water availability” with “abundant water”. - DONE

- Line 440: replace “in each” with “from each”.  - DONE

- Do not keep space between the temperature and the symbol, e.g. replace “28 °C” with “28°C”

Answer: According to international standards, such as the SI (International System of Units), there should be a space between the numerical value and the degree Celsius symbol (°C).

  1. Abstract:- The abstract is missing any actual values of measurements and names of major novel DEGs; please you need to indicate some figures to support your conclusions and interpretations!

Answer: The authors understand that figures should not be mentioned in the abstract. As requested, specific genes have been added to the abstract. Figure 5 contains a list of upregulated genes in each natural environment that correspond to the main hubs of adaptive responses.

  1. Introduction:- Language is fine.

- Relatively long, please reduce!

Answer: The introduction was also modified as requested by the other two reviewers, but the length of the introduction was not significantly reduced, as it could affect the logical flow necessary for the contextualization of the manuscript's topic.

- Figure 1 (G): it is inappropriate to show this image in highly ranked journal such as “Plants”. Please remove the lower part!

Answer: As requested, figure 1(G) was replaced by a more appropriate one.

  1. Results: - Good presentation of data.

- Very informative supplementary files.

- Two major RNA-seq data analyses are MISSING:

  1. Venn diagrams showing overlapping genes between the four major samples.

Answer: Figure 3 shows the comparative expression levels among the four sample groups. Authors consider it much more appropriate and informative than a Venn diagram as it whos the different patterns of expression among groups and not only the number of genes with common or different expression profile. Nevertheless a Venn digram was also included as a supplementary figure S1.

  1. qRT-PCR to verify the RNA-seq data.

Answer: Indeed, RT-qPCR is a very sensitive method for analyzing gene expression, but compared to direct mRNA sequencing, it is less sensitive and inappropriate for analyzing alternative splicing. This is one reason why, despite the positive correlation between the methods, their concordance does not exceed 90%. As RNA-seq was conducted with three biological replicates, the authors consider that using another method to confirm the results of such a powerful detection technique would be unnecessary.

  1. Discussion:

- Good, however incorporating more recent articles is recommended.

Answer:  References were added to the manuscript

- Please you need to discuss your data with recent similar reports such as:

- Filgueiras, J. P. C., da Silveira, T. D., Kulcheski, F. R., & Turchetto-Zolet, A. C. (2024). Unraveling the Role of MYB Transcription Factors in Abiotic Stress Responses: An Integrative Approach in Eugenia uniflora L. Plant Molecular Biology Reporter, 1-12.

- Waschburger, E. L., Guzman, F., & Turchetto-Zolet, A. C. (2022). Genome-wide identification and analysis of dof gene family in Eugenia uniflora l.(Myrtaceae). Genes, 13(12), 2235.

Answer:  These references were included in the new version of the manuscript

- You need to discuss the possible epigenetic effect from the original collection site on seedlings raised in the greenhouse!

Answer:  The authors agree that epigenetic modifications may have occurred in the progenitors present in contrasting environments and that these modifications may have been maintained in the plants evaluated under controlled conditions. To confirm such effects, experimental approaches quite different from those used in the current manuscript would be required, but they certainly need to be conducted in future studies.

- You need to discuss the around 8000 cDNAs without annotation!

Answer:  For comparison, in the genome of another Myrtaceae species available in Phytozome, there are over 5,000 CDSs associated with potential peptides that have no correlate in Arabidopsis and only potential annotation in KEGG. In the case of Eugenia uniflora, whose genome is in an even more fragmented assembly stage, this number of 8,000 may be due to this incompleteness.

  1. Materials and Methods:

- Lines 453-462: you do not need to indicate the detailed procedures as you cited the original articles.

- qRT-PCR to verify the RNA-seq data is missing.

Answer:The use of RT-qPCR to validate mRNA-seq was necessary when mRNA-seq was expensive and sample groups were limited to single or two biological replicates. However, with the inclusion of three or more biological replicates, this validation is no longer mandatory, as the statistical analyses are robust enough. Indeed, RT-qPCR is a very sensitive method for analyzing gene expression, but compared to direct mRNA sequencing, it is less sensitive and inappropriate for analyzing alternative splicing. This is one reason why, despite the positive correlation between the methods, their concordance does not exceed 90%. As RNA-seq was conducted with three biological replicates, the authors consider that using another method to confirm the results of such a powerful detection technique would be unnecessary.

  1. References:

- It is recommended to incorporate more recent articles as only 15 out 71 (ca. 21%) of cited articles were published in the last five years.

Answer:  more recent references were incorporated.